# The Influence of COVID-19 on New Lung Cancer Diagnoses, by Stage and Treatment, in Northern Italy

**DOI:** 10.3390/biology12030390

**Published:** 2023-02-28

**Authors:** Lucia Mangone, Francesco Marinelli, Isabella Bisceglia, Angelina Filice, Lisa De Leonibus, Cristian Rapicetta, Massimiliano Paci

**Affiliations:** 1Epidemiology Unit, Azienda Unità Sanitaria Locale–IRCCS di Reggio Emilia, 42122 Reggio Emilia, Italy; 2Nuclear Medicine Unit, Azienda Unità Sanitaria Locale–IRCCS di Reggio Emilia, 42122 Reggio Emilia, Italy; 3Thoracic Surgery Unit, Azienda Unità Sanitaria Locale–IRCCS di Reggio Emilia, 42122 Reggio Emilia, Italy

**Keywords:** lung cancer, COVID-19, incidence, mortality, age, stage, treatment

## Abstract

**Simple Summary:**

COVID-19 has had a dramatic impact on new cancer diagnoses and the treatment of cancer patients. Our study aimed to evaluate the impact of the pandemic on newly diagnosed lung cancer, particularly on the stage of the disease. This study documented a decrease in stage I and an increase in stage III. Additionally, a decrease in surgery was observed which was balanced by the increase in chemotherapy. Comparing the 2019–2020 trends with the previous 18 years, a decrease in the incidence in males and an increase in females was observed; mortality unfortunately follows the same trends. More than half of cancers are diagnosed at an advanced stage, so this work highlights how crucial primary prevention is, even during the pandemic.

**Abstract:**

The COVID-19 pandemic has had a significant impact on new cancer diagnoses. This study aims to evaluate the implications of the lockdown period on new lung cancer diagnoses in northern Italy. We compared 2020 with 2019 cancer registry data, reporting the variations by age, stage, and treatments. In 2020, 303 lung cancer cases were registered, 21 fewer than in 2019. Cases fell in men (−31 patients, 9.6%) but not in women (+10 patients, +3.1%). A significant drop in stage I from 19.8% to 12.9% (*p* < 0.05) and an increase in stage III (12.7% vs. 19.1%; *p* < 0.05) was observed. Histological confirmation dropped (70.1% vs. 60.1%; *p* < 0.05) while cytology increased (12% vs. 20.8%; *p* < 0.01). Surgery declined (28.7% vs. 21.5%; *p* < 0.05) but increased in stage III (19.5% vs. 25.9%; *p* = 0.46), while chemotherapy increased (17.6% vs. 34.3%; *p* < 0.01) for all stages. During the pandemic, new lung cancer diagnoses dropped only in men. The reorganization of health services has ensured a decrease in surgical interventions (due to the unavailability of operating rooms) counterbalanced by an increase in chemotherapy.

## 1. Introduction

Each year in Italy, there are about 41,000 new diagnoses of lung cancers (11% of total cases) and 34,000 deaths (19% of total deaths). Lung cancer is the second most frequent neoplasm in males over 50 and females over 70; conversely, it is the third in females over 50. Despite significant efforts, the five-year survival rate remains very low (16.4% in males and 22.7% in females); this is also true for cases diagnosed in recent years (2010–2014) [1]. The incidence trend in Italy is decreasing in males and is increasing in females [2], confirming the tendency in other Western countries.

The COVID-19 pandemic in 2020 has profoundly changed the trend in the treatment of oncological diseases both in terms of the interruption of cancer treatment, prevention, and monitoring activities and the shift of many cancer departments to COVID-19 units, with delays in diagnosis and treatment [3,4]. The first analysis, published by Liang [5], highlighted the impact of SARS-CoV-2 infections on cancer patients in China; in particular, ICU (Intensive Care Unit) admissions and deaths were higher in cancer patients, especially if cancer had been diagnosed in recent years. In the following months, many other studies were published on the same topic, emphasizing the delay in diagnosis and treatments and the impact that this has had on the incidence and mortality of patients in different sites and countries [6,7,8,9,10,11].

During the pandemic, an increased mortality risk was observed in cancer patients, especially in the elderly, males, people with comorbidities, smokers, and people with low-performance status [12]. In the UK, mortality in patients with cancer on chemotherapy is associated with higher risk, but not significantly (HR 1.5; 95% CI 0.91–2.45) [13]. The impact of chemotherapy has shown different effects [14]: in patients with lung cancer, there were higher rates of severe or critical COVID-19 events (HR 2.0; 95% CI 1.20–3.30). In addition, Italian studies showed an increased risk of being hospitalized and dying from COVID-19 compared to the general population, particularly for lung, breast, and hematological cancers [15]. However, our previous work showed an excess risk of being hospitalized for respiratory cancers (HR 3.63; 95% CI 1.26–10.44) but not of dying (HR 1.64; 95% CI 0.58–4.64) [16].

Concerning incidence, an Italian study [17] showed that during the lockdown in Italy (March–May 2020), a 45% reduction in new cancer diagnoses occurred compared with the same months of 2018–2019. In particular, the decrease concerned skin cancers and melanomas (−57%), colorectal (−47%), prostate (−45%), and lung (−27%) cancer. A subsequent study evaluated the impact that the lockdown (and the suspension of screening tests) had on new cancer diagnoses [18], highlighting a 35% decrease in new diagnoses, in particular of breast (−35%), prostate (−32%), and lung (−22%) cancer.

This work aims to describe the impact of the pandemic period on new lung cancer diagnoses in northern Italy by stage and treatment, and to compare this period with the previous 20 years.

## 2. Materials and Methods

The Reggio Emilia Cancer Registry (RE-CR) has been used from 1996 and has contributed to the collection and detailed description of oncological pathologies. The main information sources are anatomic hospital discharge records, pathology reports, and mortality data. The strengths are a high percentage of microscopic confirmation (83.5% for lung cancer) and the low rate of DCO (Death Certificate Only), below 0.1% [19]. The population covered by RE-CR amounts to 532,000 inhabitants. It was approved by the provincial Ethics Committee of Reggio Emilia on 4 August 2014 (Protocol no. 2014/0019740 of 04/08/2014). This study refers to the entire population residing in the province of Reggio Emilia at the time of diagnosis.

The availability of computerized archives and access to the oncological structures where the patients refer, allows the RE-CR staff to retrieve additional information beyond that routinely collected by CRs around the world.

Lung cancer cases were defined based on the International Classification of Diseases for Oncology, Third Edition (ICD-O-3) [20] as topography C34. This study included all non-small-cell lung cancer (NSCLC) cases diagnosed in the 2019–2020 period; we decided to exclude small-cell lung cancer (SCLC) as it would have been difficult to retrieve information on stage and treatments.

For the purposes of this study, the RE-CR staff collected information on stage (TNM 8th edition) [21], surgery, and chemotherapy by consulting the medical records in the hospital. Since this information was not routine for the CR and required additional work, we failed to collect data on radiotherapy at this time. Still, we retrieved information in general on the frequency of radiotherapy delivered in 2020 versus 2019 in patients with lung cancer. To better understand the evolution of lung cancer over the years, without focusing only on the last two years, we have extended the survey to a broader period. For this reason, the incidence and mortality trends over the last 20 years have also been reported, divided by males and females.

Descriptive analyzes of patient characteristics were performed by sex, age at diagnosis, stage, method of diagnosis, surgery, and chemotherapy, according to year of diagnosis. Age at diagnosis was divided into 4 groups: 15–54, 55–64, 65–74, and over 75. We also calculated the number of cases by stage and therapy for the two years considered. To determine the differences between the proportions in years 2019 and 2020 of these covariates, we performed a test of proportions.

For age and sex groups, specific incidence and mortality rates were calculated using the province of Reggio Emilia (recorded on 1 January of each year) as denominators. The direct method was applied to adjust rates for age and sex, using the 2013 European Standard Population as reference. We considered the last twenty years (2001–2020) for the analysis, divided into males and females.

Analyses were performed using STATA 16.1 software. In this study, we defined a *p*-value < 0.05 as statistically significant. Trends over time were analyzed by calculating the annual percent change (APC) in age-standardized rates using joinpoint regression [22]. A distribution of cases diagnosed in 2019 and 2020 by month of diagnosis has been reported in order to identify the real impact of the lockdown on new diagnoses. 

## 3. Results

In 2020, in the province of Reggio Emilia, there were 303 cases of lung cancer diagnosed (−21 cases, −6.5%) compared with 2019 (324 cases) (Table 1). Regarding age groups, slight increases were recorded in 2020 in patients under 55 and over 75. The differences by gender were more relevant: tumors decreased in males (from 212 to 181 cases, −9.6%) but not in females (from 112 to 122, +3.1%). Since the stage of the disease plays an important prognostic factor in the course of the disease, we first analyzed the differences between the cases diagnosed in 2019 and 2020 (Table 1) and then the changes that occurred in the treatments in relation to the stage of the disease (Table 2).

A significant decrease in stage I (19.8% vs. 12.9%; *p* < 0.05) and an increase in stage III (12.7% vs. 19.1%; *p* < 0.05) was observed in 2020, while stage II (7.4% in 2019 and 6.3% in 2020) and IV (58.6% in 2019 and 60.1% in 2020) remained stable. The number of unknown stage cases remained unchanged, i.e., five cases in 2019 and 2020. Clinical/instrumental confirmation (that means cases with radiological and clinical confirmation) showed no significant changes (17.9% in 2020 and 17.2% in 2019). On the other hand, histological confirmation requiring more invasive approaches decreased from 70.1% to 62% (*p* < 0.05); conversely, the less invasive cytological method increased from 12% to 20.8% (*p* < 0.01). Finally, the therapeutic attitude decreased in surgery (from 28.7% to 21.5%, *p* < 0.05) and increased in chemotherapy (from 17.6% to 34.3%, *p* < 0.01) in 2020.

A cross-evaluation between stage and treatment (Table 2) showed a slight, not significant decrease in surgery, mainly for stage I (from 89.1% to 84.6%, *p* = 0.51). A more pronounced surgical increase for stage III was detected, although still not significant (19.5% to 25.9%; *p* = 0.46). Chemotherapy doubled, for stage II tumors (from 21.8% to 42.1%; *p* = 0.16) and also increased in stage III (from 50% to 61.4%; *p* = 0.27). Notably, a significant rise in chemotherapy (from 16.4% to 33.2, *p* < 0.01) was noticed in stage IV cases.

Finally, the neoplasia incidence and mortality trends in males and females were also detected to compare the two COVID-19 years with the previous 18 years. Observing the trends for the 2001–2020 period, there was a constant decline in the incidence and mortality of NSCLC over the years in males (Figure 1a). Specifically, the incidence dropped from 83.7 cases per 100,000 p/y in 2001 to 61.9 in 2019 and, in the end, to 49.2 in 2020, while mortality declined from 73.9 cases per 100,000 p/y in 2001 to 38.4 in 2019 and later reached 38.5 in 2020. On the contrary, both incidence and mortality slightly but steadily increased in females (Figure 1b) from 21.7 cases per 100,000 p/y in 2001 to 28.7 in 2019 and 32.2 in 2020; mortality increased from 12 cases per 100,000 p/y in 2001 to 18 in 2019 and 19.8 in 2020.

Considering the last two incident years (Figure 2), the distribution of cases by month of diagnosis shows that in the three months of lockdown (March–May 2020) there was even a slight increase in diagnoses compared with the three months of the previous year. The decline in diagnoses was instead observed from September onwards, a period which in Italy coincided with the second wave of COVID-19.

## 4. Discussion

This work aimed to compare the NSCLC incidence in 2020 with that of 2019 to evaluate the impact of COVID-19 on new diagnoses and treatment. During the pandemic, Italy issued the “I stay at home” decree, with total closure or reduction in health activities and suspension of cancer screening. Since the province of Reggio Emilia is characterized by a high incidence of cancer and was strongly affected by the first wave of the pandemic of COVID-19, the first step was to evaluate the impact of the total closure of activities on new cancer diagnoses. In particular, this study investigated whether delays in diagnosis caused a shift towards more advanced forms and whether surgery and chemotherapy had strong repercussions related to the closure of non-urgent activities.

The World Health Organization officially declared the severe acute respiratory syndrome coronavirus 2 outbreak as a pandemic disease on 11 March 2020 [23]. In Italy, as above reported, the national decree completely blocked all non-emergency activities in the months of March–May 2020, a period called lockdown. However, in our province, already at the end of February 2020, schools had been closed and many activities had been suspended.

The COVID-19 infection in the world has caused various effects: apart from the deaths of citizens affected by diseases, the experience of cancer patients was even more dramatic [24,25]. This result is mainly related to two reasons, widely described in the literature in the last two years. Patients with cancer had a greater susceptibility to infection with COVID-19 and, therefore, a greater likelihood of experiencing severe adverse events and dying [26,27] from COVID-19, especially before vaccines became available. The total closure of some hospital wards and some welfare activities, the non-availability of operating rooms, and the postponement of diagnostic and therapeutic practices have further aggravated the situation [28,29]. Lung cancer, in particular, has received international attention: a large-scale retrospective study showed that patients with cancer have a 1.46 times higher risk of becoming infected with COVID-19, but people with a diagnosis in the last year, have a 7 times greater risk [30]. In particular, this study showed that the risks of infection are higher among patients with hematologic malignancies and lung cancer. In fact, among the different types of cancer, lung cancer presented different challenges since the beginning of the pandemic.

In Italy, during the three-month lockdown (March–May 2020), the closure had an impact on new cancer diagnoses, in particular breast, colorectal, prostate cancer, and melanoma [17,18]. Regarding lung cancer, the reduction was −27.4% in the Ferrara study [17] and −22% in our population-based study [18]. In our current study, observing the distribution of cases by month, we observed that in the three months of lockdown (March–May 2020) there was even a slight increase in diagnoses, compared with the three months of the last year, probably linked precisely to the high number of hospitalizations for patients with suspected COVID-19 subjected to instrumental investigations. The suspension of activities made its effects felt in the second half of the year when the so-called second wave of COVID-19 occurred in Italy, which this time caused a drop in diagnoses. There have been many studies that have documented the outcomes of lung cancer during the pandemic: an increase in patients infected by SARS-CoV-2, an increase in ICU admission and the use of artificial ventilation, and a 25–30% increase in the probability of dying [31], in particular in smokers (HR 2.9; 95% CI 1.07–9.44) or patients with previous obstructive events (HR 3.87; 95% CI 1.35–9.68). These outcomes are related mainly to the delays in diagnostic tests and treatments [32], with particular emphasis on mono-CT treatments [33]. Sometimes, the delays were associated with the patient’s fear of COVID-19 rather than the tumor itself [34], even if the fear of cancer was prevalent in different contexts [35]. The dramatic impact that COVID-19 infection can have on lung cancer patients has highlighted the need to implement infection control measures and reorganize cancer management care [36]. Data on the potential interference between cancer patients and SARS-CoV-2 infection are still inconclusive. Consequently, oncologists have had to find the balance between treating patients and protecting them from SARS-CoV-2 infection [37]. Where present, there was an interruption of lung screening programs with a drop in the diagnosis of nodules highly suspected for cancer (from 29% to 8%) [38]. Certainly, the reorganization of health structures has meant that in some contexts, just 10% of patients who needed ICU care were admitted for treatment [39]. Lung cancer patients have some special characteristics compared to other cancer patients: they often have comorbidities and pre-existing diseases, which may present increased risk of infection; but on the other hand, since the associated symptoms (such as cough, dyspnea, and hemoptysis) are the same as for COVID-19 [40,41,42], this increased the possibility of being subjected to radiological investigations even during the pandemic and discovering lung tumors [43].

But in addition to the impact on hospitalizations and mortality, the question arises with a decline in new cancer diagnoses. Due to the overlap of SARS-CoV-2 symptoms with those of lung cancer [44], new lung cancer diagnoses were accounted for in many of our patients. A Canadian study [45] showed that in 2020 there were 103 cases, compared to 130 diagnosed in 2019 (−34.7%), which is higher than the number registered in our study (6.5%). Since lung cancer survival has remained very low in recent years, early diagnosis is crucial for this neoplasm [46]. One of the effects of the pandemic has been to close many healthcare activities (screening, outpatient visits, follow-up visits, etc.) and since the high number of patients with COVID-19 required hospitalization, the redistribution of human resources has become a priority for many institutions [47]. Studying the impact of COVID-19 on new cancer diagnoses requires updated data from cancer registries. For example, the Netherlands National Cancer Registry showed a 25% drop in diagnoses between March and May 2020 compared to the previous year [8]. A study by Park [48] analyzed the impact of COVID-19 on lung cancer in Korea: the proportion of patients with stage III and IV NSCLC increased from 62.7% in 2019 to 74.7% in 2020 (in our study from 71.3% to 79.2%). These data suggest the existence of a delay in NSCLC diagnosis during the COVID-19 pandemic [48]. A retrospective study [49] analyzed lung cancers before and after the UK lockdown period: the authors reported a 7.6% increase in stage III and a decrease in stage II. In our work we also recorded an increase in stage III of 6.4%, associated with a decline in stage I. Reyes [50] showed that in the period January–June 2020 vs. 2019 there was a 38% drop in diagnoses. Unfortunately, it has been estimated that delayed diagnosis due to COVID-19 could cause more than 1000 lung cancer deaths within 5 years of diagnosis [49].

In our study, we observed a decrease in early forms (from 19.8% to 12.9%; *p* < 0.05 for stage I), while both advanced forms (from 12.7% to 19.1%; *p* < 0.05 for stage III) and metastatic tumors (from 58.6% to 60.1%; *p* = 0.72 for stage IV), increased. The observed results are not surprising and are in line with the literature. What is surprising, however, is the fact that surgery has declined, but not for stage III: in fact, the indications were to ensure surgical treatment for these patients, when required. This is in line with what is reported in the literature: stage III represents a high priority since a delay could cause a shift towards more aggressive forms that are no longer operable [51,52].

Regarding treatments, Kasymjanova (who also included small-cell lung cancer) showed a decrease in therapies (85% vs. 82%), in particular surgery drops (38% vs. 25%; *p* = 0.09) and a slight decline in chemotherapy (from 37% to 30%; *p* = 0.07). The increase in radiotherapy [45] has counterbalanced the reduction in surgery and chemotherapy. Also, our study observed a decrease in surgery, offset by an increase in chemotherapy and a 20% increase in radiotherapy delivered in lung cancer patients in 2020 compared to 2019.

In general, patients’ access to the hospital was limited if strictly necessary [44]. On the other hand, guaranteeing cancer treatments was a priority since delays in treatment could cause an increase in morbidity or mortality. Given this situation, since the lockdown, the primary international oncology societies, including the American Society of Clinical Oncology (ASCO), the European Society for Medical Oncology (ESMO), and the Italian Association of Medical Oncology (AIOM), have proposed a variety of recommendations to help oncologists to guarantee the continuation of cancer treatments safely and to ensure the best standard of care [53,54], based on epidemic area, local healthcare structure capacity, the individual risk of infection, stage of disease, patient comorbidities, age, and treatment characteristics [55].

The COVID-19 pandemic revealed the need for more information and guidance for changes in existing practice among lung cancer specialists, including thoracic surgeons. Many institutions have changed the treatment plan to minimize the risk of patient exposure, by recommendations of certain expert groups [51,56,57,58].

However, in our context, it appears that a reasonable health organization has managed to “contain” the restrictive measures imposed upon the population by the pandemic. Pleural mesothelioma also confirmed this result. In Italy, the diagnoses dropped in the south, which was almost unaffected in the initial pandemic phases, while was maintained in the north, demonstrating the organization’s resilience despite the pandemic [59].

Among the strengths of this study, we would include the quality data, guaranteed by the high percentage of microscopic confirmations and the low percentage of DCO; the availability of 20 years of incidence data which enables the observation of what happened during the pandemic with greater objectivity; and the availability of population data, which make it possible to overcome selection biases related to only inclusion of hospital cases. The most important strength of this work is the availability of incident data up to 2020 (no CR in Italy has published population data) and the collection of information on stage and treatment (which are not variables routinely collected by CRs around the world). Since cancer registration takes place in an active form (registrars must actively search for information), this represented a huge effort for our CR.

This study has some significant limitations: There is a lack of information on radiotherapy, which is also widely used in lung cancer, and its use was probably completely different during the pandemic. Since radiotherapy is not reported in the available databases, it was impossible to retrieve this information from the individual patient level. Furthermore, information on the comorbidities that strongly affect lung cancer treatments and the prognosis of the disease is lacking. Finally, the data presented refer to a single CR, thus representing a limit since the results cannot be transferred to the areas of northern Italy equally distressed by a high incidence of tumors and deeply affected by the pandemic.

Nevertheless, this is the only CR in Italy with data updated to 2020 and was, therefore, able to offer an overview of the pre- and post-COVID-19 situation. The availability of data from the remaining CRs will allow for a better interpretation of some results and, above all, to compare different areas and populations. Even the analysis of pre- and post- pandemic survival could add useful information for research and clinical practice.

## 5. Conclusions

The pandemic confirmed the increasing trend of lung cancer, but only in women. New lung cancer diagnoses dropped during the pandemic, but only in men. In general, we observed a decrease in early cancers and an increase in stage III. However, surgery was ensured for stage III patients who represented a priority. The total closure of activities has caused a slight increase in lung cancer diagnoses (with symptoms similar to those of COVID-19) but the second wave of infections in Italy has caused a sharp decline in new diagnoses. The first results of the work were shared with doctors and healthcare professionals in the context of a hospital meeting with the aim of understanding where the organizational deficiencies were and where urgent action was needed to avoid negative consequences for patients. As more than half of lung cancer cases continue to be diagnosed in the advanced stage, primary prevention (cessation of cigarette smoking) must still remain a powerful focus on which to direct our efforts.

## Figures and Tables

**Figure 1 biology-12-00390-f001:**
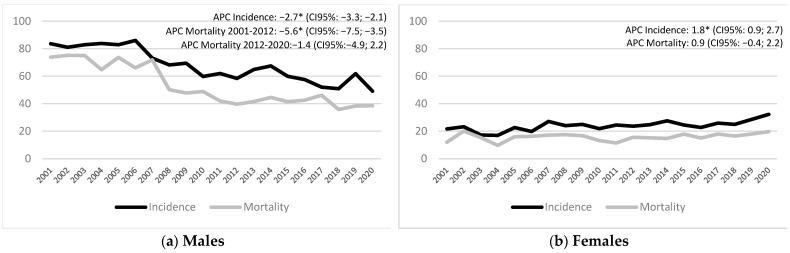
Incidence and mortality trend of lung cancer in males (a) and females (b). Years: 2001–2020. * APC is significantly different from zero at the α = 0.05 level.

**Figure 2 biology-12-00390-f002:**
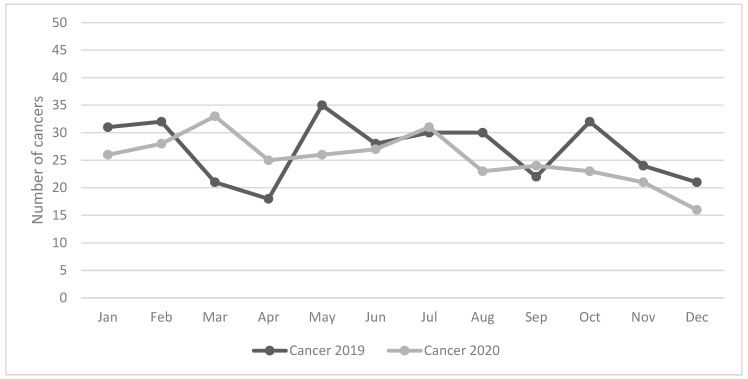
Number of cases of lung cancer by month of diagnosis. Years: 2019 and 2020.

**Table 1 biology-12-00390-t001:** Number of cases by age, sex, stage, method of diagnosis, and therapy, per year.

	Years					Total (n = 627)	
	2019 (n = 324)		2020 (n = 303)				
	n	%	n	%	*p*-Value	n	%
Age at diagnosis							
15–54	16	4.9	20	6.6	0.37	36	5.7
55–64	62	19.2	53	17.5	0.59	115	18.3
65–74	119	36.7	108	35.6	0.78	227	36.2
75+	127	39.2	122	40.3	0.79	249	39.7
Sex							
Males	212	65.4	181	59.7	0.14	393	62.7
Females	112	34.6	122	40.3	0.14	234	37.3
Stage							
I	64	19.8	39	12.9	<0.05	103	16.4
II	24	7.4	19	6.3	0.57	43	6.9
III	41	12.7	58	19.1	<0.05	99	15.8
IV	190	58.6	182	60.1	0.72	372	59.3
**Unknown**	5	1.5	5	1.6	0.91	10	1.6
Method of diagnosis							
Histological	227	70.1	188	62.0	<0.05	415	66.2
Cytological	39	12.0	63	20.8	<0.01	102	16.3
Clinical/instrumental	58	17.9	52	17.2	0.81	110	17.5
Surgery							
Yes	93	28.7	65	21.5	<0.05	158	25.2
No	231	71.3	238	78.5	<0.05	469	74.8
Chemotherapy							
Yes	57	17.6	104	34.3	<0.01	161	25.7
No	258	79.6	195	64.4	<0.01	453	72.3
Unknown	9	2.8	4	1.3	0.20	13	2.1

**Table 2 biology-12-00390-t002:** Number of cases by stage and therapy per year.

	Surgery					Chemotherapy				
	2019		2020			2019		2020		
	n	%*	n	%*	*p*-Value	n	% *	n	% *	*p*-Value
Stage										
I	57	89.1	33	84.6	0.51	1	1.6	1	2.7	0.71
II	19	79.2	15	79.0	0.99	5	21.8	8	42.1	0.16
III	8	19.5	15	25.9	0.46	20	50.0	35	61.4	0.27
IV	6	3.2	2	1.1	0.17	31	16.4	60	33.2	<0.01
Unknown	3	60.0	0	0.0	<0.05	0	0.0	0	0.0	/

* Percentages were calculated considering the total number of single-stage cancers per year.

## Data Availability

The data presented in this study are available on request from the corresponding author. The data are not publicly available due to ethical and privacy issues; requests for data must be approved by the Ethics Committee after the presentation of a study protocol.

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
