# Peer review of "The Influence of COVID-19 on New Lung Cancer Diagnoses, by Stage and Treatment, in Northern Italy"

_biology, 2023, doi:10.3390/biology12030390_

Round 1
Reviewer 1 Report
There are several weaknesses in this manuscript. First, it aims to investigate the influences of COVID-19 on new cases of lung cancer diagnosis, but only 2020 data was included. With the comparison of data in 2019 and 2020, the results were weak. Authors should include data in 2021-2022, and 2023 if possible, to report the findings with more solid data. Second, there are so many confounding factors in Table 1, the authors should use statistical methods to identify the driving factors amongst all possible factors. Third, the explanation for stages was vague, please explain explicitly for what is diagnosis by stage, and what does it mean for patients and treatments. Additionally, in line 191-198, the authors pointed out the importance of different periods: pre-lock down, and three-month lockdown, but there was no analysis for different periods. The implications for future study were very weak, since only with data in 2020, the conclusions for post-COVID cannot be made.
Author Response
Thanks for the comments, we hope that our responses are compliant with your requests.
Best regards,
Lucia Mangone

Reviewer 2 Report
I read, The Covid-19 influence on new lung cancer diagnosis by stage, with interest. In this manuscript, the authors aimed to describe the impact of the pandemic period on new lung cancer diagnoses by stage and treatment and to compare this period with the previous 20 years.
I have some questions and suggestion.
1) Can you explain why this study is new or telling new things?
2) Discussions in the first paragraph should include a brief summary of the study's findings.
3) Discussion is rather weak. The data from other studies is relatively small. Please add more data about the impact of the pandemic period on new cancer diagnoses from the other studies as well for comparison with this study.
4) Please provide more data on the importance of physicians, and healthcare professionals around the world recognizing the pandemic period on new lung cancer diagnoses.
Author Response
Thanks for your comments. We hope that our responses are compliant with your requests.
Best regards,
Lucia Mangone

Round 2
Reviewer 1 Report
The manuscript compared the differences in the lung cancer diagnosis by age, sex, stage, method of diagnosis, surgery, and chemotherapy between 2019 and 2020 to investigate the influence of Covid-19. Although the findings can be potentially important, the reviewer thinks that the methodology section is unclear and that the discussion section needs improvement.
Here are some detailed comments:
1. What is the population covered by RE-CR or covered by the current study? (Are they inhabitants of Reggio Emilia Province, Italy?) The scope of the current study should be clearly pointed out in the title, the last paragraph of Introduction, Materials and Methods, and Results.
2. It is unclear to the reviewer how the one-way ANOVA test was performed to compare the differences between 2019 and 2020. Take the comparison of age groups for example. The ratio of age 15-54 in 2019 is 4.9%, and the ratio of age 15-54 in 2020 is 6.6%. How was ANOVA performed on only two values?
3. The authors did well in adding comparisons with similar studies in the Discussion section. However, some background information such as cigarette smoking and lockdown period should be placed in Introduction or Materials and Methods instead of Discussion. Also, the Discussion section can be better organized to make it easier to follow.
Author Response
Thanks for the comments, we hope we have answered exhaustively.
Best regards,
Lucia Mangone

Round 3
Reviewer 1 Report
The comments have been addressed accordingly.